# Outdoor recreational activity experiences improve psychological wellbeing of military veterans with post-traumatic stress disorder: Positive findings from a pilot study and a randomised controlled trial

Mark Wheeler[1], Nicholas R. Cooper [1]*, Leanne Andrews[2], Jamie Hacker Hughes[3], Marie Juanchich[1], Tim Rakow[4], Sheina Orbell[1]

1 Department of Psychology, University of Essex, Colchester, Essex, United Kingdom, 2 Department of Health and Social Care, University of Essex, Colchester, Essex, United Kingdom, 3 Northern Hub for Veterans and Families Research, University of Northumbria, Newcastle upon Tyne, United Kingdom, 4 Department of Psychology, Institute of Psychiatry, Psychology and Neuroscience; King's College, London, United Kingdom

* ncooper@essex.ac.uk

## Abstract

Exposure to the natural environment is increasingly considered to benefit psychological health. Recent reports in the literature also suggest that outdoor exposure that includes recreational pursuits such as surfing or fishing coupled with opportunities for social interaction with peers may be beneficial to Armed Forces Veterans experiencing Post-Traumatic Stress Disorder (PTSD). Two studies were conducted to evaluate this possibility. In particular, these studies aimed to test the hypothesis that a brief group outdoor activity would decrease participants' symptoms as assessed by established measures of PTSD, depression, anxiety and perceived stress, and increase participants' sense of general social functioning and psychological growth. Experiment one employed a repeated measures design in which UK men and women military veterans with PTSD (N = 30) participated in a group outdoor activity (angling, equine care, or archery and falconry combined). Psychological measures were taken at 2 weeks prior, 2 weeks post, and at 4 month follow up. We obtained a significant within participant main effect indicating significant reduction in PTSD symptoms. Experiment two was a waitlist controlled randomised experiment employing an angling experience (N = 18) and 2 week follow up. In experiment 2 the predicted interaction of Group (Experimental vs. Waitlist Control) X Time (2 weeks pre vs. 2 weeks post) was obtained indicating that the experience resulted in significant reduction in PTSD symptoms relative to waitlist controls. The effect size was large. Additional analyses confirmed that the observed effects might also be considered clinically significant and reliable. In sum, peer outdoor experiences are beneficial and offer potential to complement existing provision for military veterans with Post Traumatic Stress Disorder.

**Data Availability Statement:** All data files are available from the OSFHOME database (https://osf.io/63hrb/?view_only= 2c8ffabdc8a64e77ba649c47f6df6c75).

**Funding:** The author(s) received no specific funding for this work.

**Competing interests:** I have read the journal's policy and the authors of this manuscript have the following competing interests: Following the results of the studies reported in this manuscript (and subsequent work), two authors (Wheeler & Cooper) have set up a Community Interest Company (iCARP CIC) in order to provide more outdoor pursuit experiences for more veterans. They receive no income for this venture. This does not alter our adherence to PLOS ONE policies on sharing data and materials.

## Trial registration

The authors confirm that all ongoing and related trials for this intervention are registered.

The studies reported in this manuscript are registered as clinical trials at ISRCTN:

Pilot ID– ISRCTN15325073

RCT ID– ISRCTN59395217

## Introduction

Post-traumatic stress disorder (PTSD: DSM-5 [1]) is a disabling psychological condition comprising four main sets of symptoms: re-experiencing, hyperarousal, avoidance and negative changes in thoughts and mood. Re-experiencing refers to intrusive thoughts or images, flashbacks and/or nightmares. Hyper-arousal may be characterised by sleep disturbance, irritability, anger and hyper-vigilance for threat in the environment. People with PTSD tend to cope by avoidance of situations and social interactions and have difficulty in regulating emotions. Social maladjustment, phobia, anger, violent behaviour and family discord are often associated with combat-related PTSD [2]. PTSD is also frequently comorbid with other mental health problems such as depression, anxiety, substance abuse and suicidal ideation [3, 4]. If left untreated, PTSD may become a chronically disabling condition associated with impaired occupational, relational and social functioning [5].

It is notoriously difficult to estimate the numbers of military veterans with PTSD. A comprehensive review of evidence from multiple sources [6] suggests a prevalence of between 2–17% amongst US veterans and 3–6% of UK veterans, with estimated rates affected by a range of methodological factors such as sampling strategy, measures, criteria employed to determine cases, as well as combat role and cultural background. Given that, in the UK for example, there are more than one million military veterans aged 20–69 [7], and in the US approximately 20.1 million [8], these percentages translate into substantial numbers of men and women living with a condition that impacts upon their wellbeing and life quality, capacity to work, maintain personal and parental relationships and social connectedness, all of which incur substantial personal and public costs. While established evidence based psychological treatments exist for PTSD [e.g. 9], evidence suggests that military veterans with PTSD in particular do not benefit adequately from such provision. The present paper reports upon the rationale, development and evaluation of a supplemental outdoor recreational experience approach to reach UK military veterans with PTSD.

### Limitations of existing treatment for military veterans with PTSD

Studies have found that large proportions of veterans with mental health problems do not receive any mental health treatment [10–12]. Hoge et al. [10] report that only 23–40% of veterans with health disorders had sought treatment. Kuehn [11] reports that more than 45% of US military veterans with PTSD who were referred for treatment never received any form of therapy. There is also evidence of significant delay in treatment seeking; Murphy [3] reports that veterans may experience mental health difficulties for as long as 12 years after leaving service before seeking help. Literature indicates that three significant issues impact upon the ability of military veterans to benefit from standard provision of treatment for PTSD. These relate to barriers to enter or remain engaged in treatment, reduced effectiveness of treatment and comorbidity of PTSD with other conditions.

Barriers to enter treatment have been extensively documented and include beliefs about mental health treatments, perceived stigma and access barriers [13–17]. Mellotte et al. [18] distinguished between barriers to *enter treatment* and barriers *to progression through treatment* in UK veterans. Barriers to treatment entry included minimizing or not recognizing that symptoms were psychological as opposed to physical, shame and embarrassment, anticipated negative judgements from others, including fear of being perceived as weak or malingering by civilians. Barriers to *progression* through treatment once initiated were more concerned with service delivery; lengthy waiting times, difficulties with coping with busy public transport services, and health professionals who lacked necessary military specific knowledge and terminology. All participants who sought help did so only when they reached a crisis and risked losing their life, liberty, family or job. There is also evidence that established psychological therapies may be less effective and terminated prematurely by military veterans [19, 20]. In their meta analytic review of effects of cognitive behaviour therapy and eye movement desensitization in treatment of PTSD, Bradley et al. [21] obtained an aggregate effect size for treatment of combat trauma that was less than half that obtained in the treatment of other traumas. The authors suggest that one important factor is the tendency to limit disclosure at home (to civilians) so that engagement in 'homework' is limited and participants consequently do not avail of important social support during therapy. Comorbidity also complicates treatment [3, 4, 22] and extreme avoidance, anger reactions and loss of hope may also contribute to early departures from treatment. In sum, various lines of evidence point to the need for innovative approaches to overcome the barriers to mental health care amongst military veterans.

## An alternative approach: Outdoor recreation experiences

A developing body of evidence points to the positive impact of exposure to natural environments on psychological wellbeing in general populations. Several reviews [e.g. 23–26] suggest that exposure to nature and outdoor recreation can improve attention and cognition, memory, stress and anxiety, sleep and quality of life. Natural environment exposure is thought to benefit wellbeing via a number of mechanisms; at the forefront of these is attention restoration theory [27]. Attention restoration theory suggests that one's environment can influence cognition and behaviour in terms of workload and induced fatigue. Specifically, urban environments require directed attention and increased processing of stimuli, thereby increasing cognitive load and fatigue. In contrast, natural environments require less directed attention, eliciting more putative 'soft fascination' compared to the 'hard fascination' that tends to occur in urban environments. This soft fascination allows involuntary attention and aids recovery from fatigue.

In recent years researchers have begun to explore the possibility that outdoor recreational activity experiences may have therapeutic benefit to military veterans with PTSD. Outdoor recreational experiences have attracted a multitude of labels including 'green exercise' [23, 28], 'therapeutic recreation' [29], 'forest bathing' [30] 'peer outdoor support therapy' [31], 'nature adventure rehabilitation' [32], 'nature based therapy' [33] and 'nature recreation experience' [34]. These interventions often combine the benefits of a natural environment with learning a new recreational skill from certified professionals.

Greer and Vin-Raviv [35] identified 13 articles, all but one published since 2011, providing a quantitative (n = 9) or qualitative (n = 4) evaluation of some form of outdoor recreation-based intervention offered to military veterans with PTSD. The majority were conducted in the USA, one in Israel and one in Denmark. The type of recreation offered by professional instructors included horticulture, sailing, fly-fishing, surfing, hiking and snow sports delivered to small groups of veterans, and ranging in duration and intensity from two days [e.g. 36] to

once weekly for twelve months [32, 36]. The focus of these interventions, rather than being the active treatment of PTSD, is on learning a new recreational skill, thereby avoiding elements of shame or stigma and known barriers to therapy in this sub-population. In learning a new skill, participants may be distracted from everyday concerns whilst engaged in the task at hand and have an opportunity to practise problem solving, and solution rather than avoidant coping modes to overcome difficulties. This is important because chronic PTSD clients often minimize their interactions with the environment and with people, rarely encountering real life challenges [32]. Success and enjoyment in learning a new skill may address hope and facilitate development of a sense of identity and purpose beyond PTSD and the military. Frequent breaks between bouts of activity and passivity may foster experience of natural cycles of emotional regulation. Importantly, the natural environment provides a break from constant hypervigilance and reactivity to sudden sensory inputs and induces relaxation [33, 37]. It has also been suggested that natural environments *per se*, may represent calm and familiarity to veterans because of the many hours spent training in natural environments [38]. A third important element of these interventions is that they involve small groups of military veterans with PTSD. Peer groups have been observed to have great therapeutic potential [31]. Groups of veterans have the potential to recreate an 'esprit de corps' that strengthens belonging and may develop a social network that facilitates sharing of tips such as how to handle particular situations [32]. Mellotte and Murphy [18] observed that discomfort when speaking to civilians was a factor in discontinuation of treatment. Peer groups may also facilitate a feeling of safety as a consequence of being amongst veterans facing common challenges, who share a common language and understanding. In sum, outdoor therapeutic recreation comprises three important elements; being in an outdoor natural environment, being amongst other veterans and professional instruction in a new recreational activity.

Evidence from this evolving literature to date suggests that interventions that incorporate these three elements have had some success in demonstrating statistically significant pre- to post-intervention changes in PTSD symptomology [29, 32, 36, 39–41] and depression [29, 32, 36, 39, 41, 42]. However, few studies followed up post-intervention [e.g. 36] and some studies found that improvements were not sustained post intervention [e.g. 29, 41]. To date, only two studies have employed a comparison group. Hyer [43] employed a quasi-experimental design in an evaluation of a 5-day outward bound experience including a range of activities including ropes, climbing, hiking and white water rafting. No difference was obtained between the intervention and comparison group at the end of the intervention. Gelkopf [32] evaluated a 12-month programme involving 3 hours of sailing per week. Participants randomly allocated to the intervention or waitlist control group were significantly different in PTSD symptoms, daily functioning, hope and depression at the end of the programme.

## The present studies

The present studies added to the developing literature in outdoor therapeutic recreation for military veterans, by conducting studies amongst British military veterans with PTSD. These studies extend previous literature in a number of respects. We compared three different outdoor recreational activities that were available locally. Previous research has explored a multitude of recreational activities, many provided in rather exotic and stunning locations far from participants' homes, such as Green River Utah or the Appalachian Trail [e.g. 36, 40, 42–44]. We developed and tested an intervention that could be delivered close to veterans' homes and represent minimal practical access difficulties to veterans. This intervention could be readily delivered in many locations in many parts of the world. We sought to discover if these activities and contexts might produce results similar to those obtained in more dramatic natural

environments. Moreover, we considered that a locally delivered intervention might enhance possibilities for subsequent social support from other veterans taking part. We also sought to address some limitations of previous research. First, we excluded all potential participants who were in receipt of psychotherapy. Military veterans often avoid psychotherapeutic treatment for many years, and as noted by Gelpkof [32], interventions such as these may be particularly valuable for those not yet ready to engage, who have not benefitted or who have been treatment drop-outs. Also, the inclusion of participants who were concurrently receiving psychotherapy in some previous studies [e.g. 40] makes it difficult to discern if benefit arose from the novel intervention or from ongoing psychotherapy. Second, only two previous studies have attempted a control comparison [35]. Consequently, after estimating the effect sizes obtained in our first experiment we conducted a power analysis and conducted a controlled experiment with random allocation to experimental condition in our second experiment.

## Aims

We designed two experiments to contribute to and extend previous literature by providing an evaluation of the effects of brief peer group outdoor recreational activity experiences and the potential for military veterans to engage in such interventions. The two experiments employed different samples of military veterans with PTSD in the UK. The experimental interventions were targeted at military veterans with PTSD diagnoses who were not in receipt of psychological therapy. An outdoor recreational experience was provided and led by professional coaches. A procedure was developed for the delivery of the intervention to ensure replicability. The first experiment compared three different types of outdoor experience (angling, equine care, falconry/archery) and employed a within participant design with follow up at 2 weeks and 4 months. It was hypothesized that participants would experience a decrease in symptoms as assessed by established measures of PTSD, depression, anxiety and perceived stress as a consequence of the experience. Encouraged by the findings of experiment one, we conducted a wait-list-controlled experiment (angling: experiment two) to compare the effect of the experience with a control group. The hypotheses that experimental participants would experience a decrease in symptoms of PTSD, depression, anxiety and perceived stress and an improvement in ratings of general social functioning and psychological growth, relative to controls, were supported.

## Ethical approval statement

The University faculty ethical review committee granted approval for the experiments. A health and safety risk assessment was also completed and fully trained professional coaches and a high intensity psychological therapist were present during the experiments. Participants were volunteers who gave written consent to take part in the experiments. They were informed that all co-participants would be military veterans with the same diagnosis and that a mental health professional was on site as well as professional angling/riding/falconry coaches. The authors confirm that all ongoing and related trials for this intervention are registered (IDs: ISRCTN15325073 and ISRCTN59395217).

## Experiment one

### Method

The studies reported in this manuscript were approved by the University of Essex Ethics Committee (ethics ID: MW1501/2). All participants gave informed written consent before taking part in the studies.

**Design and participants.** The experiment employed a pretest-posttest within participant design (time: pre-intervention, 2 weeks post-intervention, 4 months post-intervention) with one between groups factor (type of activity: angling, equine, falconry and archery combined). Each activity intervention was designed to deliver an outdoor recreational activity in a peer group context and to facilitate opportunities to socialise and to discuss military experience or PTSD experience if the participant so wished. The three different activity interventions ran sequentially and employed the same eligibility criteria, recruitment process and evaluation.

The experiments took place in the environs of a super-garrison town in the UK. Participants were recruited from a population of 65 service users registered at a local military welfare service (Veterans First). The service provides a social coffee morning where veterans and their families can meet up. There are representatives from military charities in attendance, who can address any questions the veterans may have. The service consented 30 willing volunteers (25 men and 5 women) who met the eligibility criteria: military veteran with a formal diagnosis of PTSD by a National Health Service or Ministry of Defence psychiatrist. Diagnosis was confirmed through documents showing a formal diagnosis by a psychiatrist presented by the participants to Veterans First. Definition of the term military veteran differs between countries. In the UK, the term 'military veteran' applies to anyone person "who has performed military service for at least one day and drawn a day's pay" [45; pg. 2]; however, all our participants had served in the military for considerably longer (mean length of 11 years (SD 6.12)). No participant was currently receiving psychological therapy for PTSD. Participants were randomly allocated (using an online randomisation tool– www.random.org) to either an angling, equine husbandry or falconry and archery combined recreational experience.

A summary of participant characteristics (including service length) is shown in the left-hand portion of Table 1. The majority of participants were unemployed, taking prescribed psychotropic mediation and had left military service on average 11 years previously. Participants were informed that they would be provided with the opportunity to learn a new recreational activity from qualified specialists, connect with other veterans with PTSD who share similar life challenges, and enjoy the setting. Recruitment began in August 2014 and data collection continued until June 2015 (4-month follow-up). All screening and data collection was carried out via telephone interview by a research assistant.

**Intervention description.** Each intervention was designed to deliver a day-long outdoor recreational experience involving tuition in a peer group context. Attention was given to creating opportunities for participants to interact with each other. In each context, the venue was made available exclusively to the veterans for the duration of the experience. Professional coaches (in angling, horse husbandry and riding, falconry and archery) provided instruction and were available at a ratio of two participants to one coach. A description of the coaching provision and activities undertaken in each intervention can be seen in Table 2 below.

**Table 1. Summary of participant characteristics, Experiments 1 and 2.**

| | Experiment One | | | | Experiment Two | | |
|---|---|---|---|---|---|---|---|
| | **Total** | **Angling** | **Equine** | **Falconry** | **Total** | **Intervention** | **Wait List** |
| | **N = 30** | **N = 11** | **N = 8** | **N = 11** | **N = 18** | **N = 9** | **N = 9** |
| No. (%) men | 25 (83%) | 10 | 5 | 10 | 17 (94%) | 8 | 9 |
| Mean age (SD) | 42.3 (9.1) | 38 (8.9) | 41 (7.8) | 48 (10.7) | 40.00 (12.70) | 41 (13.47) | 38 (12.56) |
| Years' military Service (SD) | 11.0 (6.12) | 9.18 (4.45) | 10.06 (6.62) | 13.77 (7.30) | 10.06 (5.33) | 9.33 (4.06) | 10.78 (6.53) |
| Years since leaving service (SD) | 11.04 (9.47) | 8.36 (8.04) | 10.13 (9.34) | 14.64 (11.04) | 12.39 (10.84) | 14.11 (10.98) | 10.67 (11.06) |
| N (%) not employed | 22 (73%) | 5 | 8 | 9 | 8 (44%) | 4 | 4 |
| N (%) taking psychotropic medication | 24 (80%) | 8 | 7 | 9 | 14 (78%) | 7 | 7 |

**Table 2. Coaching provision and idiosyncratic and common activities for all three interventions in Experiment 1 (see text below for common activities).**

| Intervention | Coach Provision | Distinctive Activities | Common Activities |
|---|---|---|---|
| Equine Husbandry | 5 riding instructors supplied by the stables | Mucking out stables, feeding and grooming of horses. Horseback riding led by instructors | Minibus transportation. |
| | | | Health and safety briefing. |
| | | | Collaborative creation of a communal area for food and hot drink preparation by the veterans. |
| | | | Communal meals. |
| Falconry & Archery | 4 raptor handlers and qualified archery coaches | Flying the birds to glove, feeding birds, cleaning aviaries. Target practice, then competition in archery | Minibus transportation. |
| | | | Health and safety briefing. |
| | | | Collaborative creation of a communal area for food and hot drink preparation by the veterans. |
| | | | Communal meals. |
| Angling | 5 professional anglers were provided by the bait company some of whom also had level 1 or 2 angling coach certification from the Angling Trust, UK | Learning of angling skills and techniques, including mastery of rig tying, bait application and fish care | Minibus transportation. |
| | | | Health and safety briefing. |
| | | | Collaborative creation of a communal area for food and hot drink preparation by the veterans. |
| | | | Communal meals. |

In addition to the different activities pursued, each outdoor activity experience contained the same common elements. Participants were transported to the venue by minibus. On arrival at the venue, a health and safety briefing took place. Participants were then allocated to coaches and provided with equipment (and designated horse in the case of equine, and designated fishing spot around a lake in the case of angling). Participants collaborated in setting up a communal area for the purpose of socialising, eating and taking warm drink breaks. Food (e.g. sausages, burgers, chicken, salad etc.) was provided to be prepared, cooked and shared by participants communally. The focus was on the recreational activity led by qualified coaches in a natural environment alongside veteran peers. At the end of the experience participants were encouraged to create a 'Facebook' group in order to keep in contact via social media. At the end of the day, participants were transported home by minibus.

A qualified mental health professional was on site throughout to observe and monitor signs of distress, and if necessary to assist any participant who experienced flashbacks during the experience, but no formal psychological therapy was offered or delivered during the intervention and there was no deliberate initiation of discussions relating to trauma. The mental health professional did *respond to* questions about PTSD and provided some basic information and signposting to appropriate services if approached.

The angling context provided participants with tents and tackle situated around a lake. Participants were free to move around the lake and talk to other participants. The equine context involved participants collaborating in pairs to groom, 'muck out', prepare food and bedding and clean tack for their own horses. They were then taught riding skills before embarking on a horseback walk in surrounding fields. The falconry and archery context provided participants with a half-day falconry and a half-day archery in two groups of 5 and 4 who swapped at lunchtime. Participants learned how to handle and fly raptors and were coached in archery.

**Measures.** Repeated measures were taken at three time points: two weeks prior to intervention(baseline), two weeks post intervention, and four months post intervention (see Fig 1). The researcher collected all measures by telephone. Four established and validated measures were included to assess mental health. PTSD symptoms were assessed by the PTSD Checklist Military (PCL-M; [46]). The Posttraumatic Stress Disorder Checklist is a commonly used

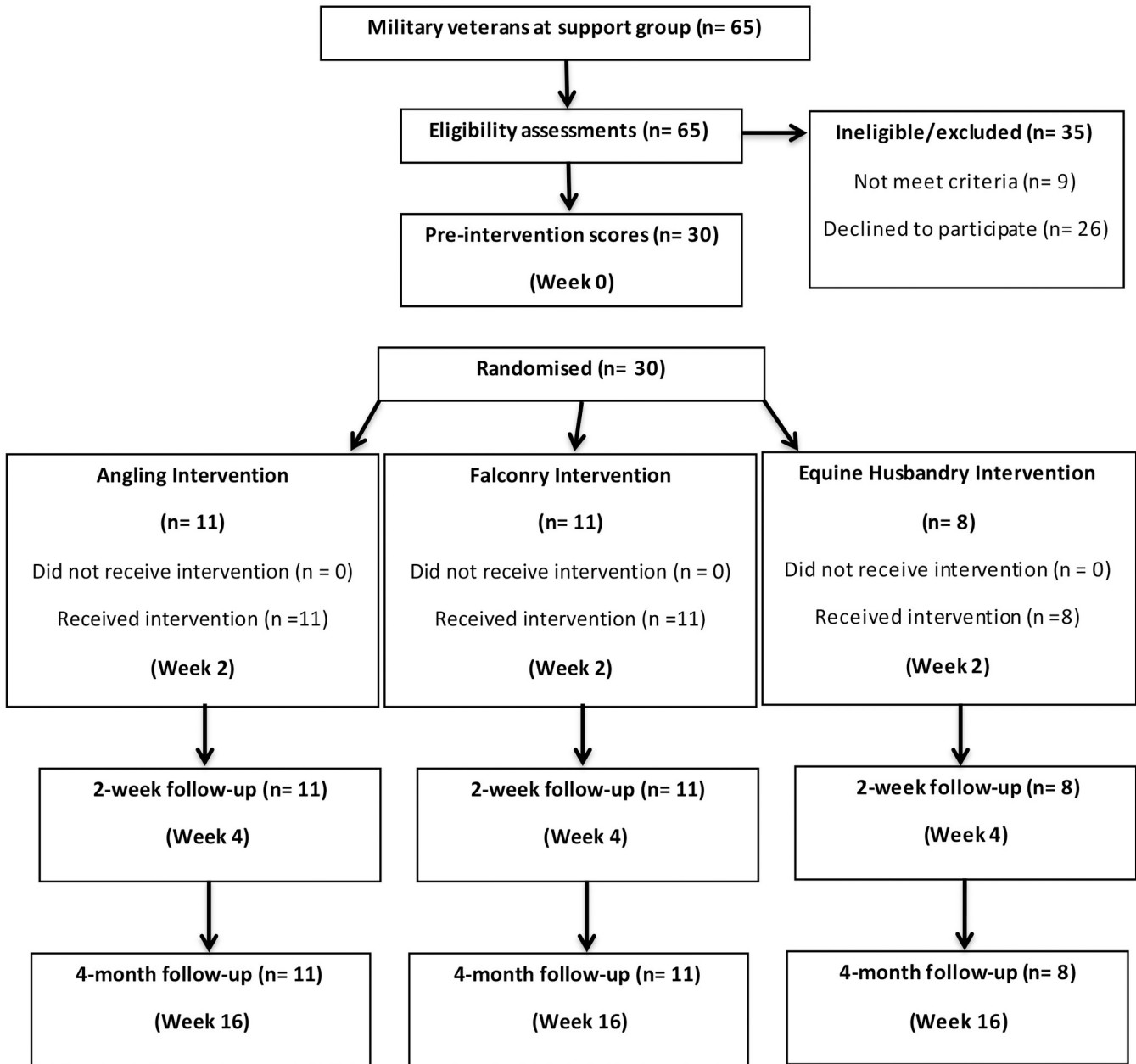

**Fig 1. Summary of measurement and intervention timepoints for angling, equine husbandry and archery/falconry groups.**

measure, with military (PCL-M), civilian (PCL-C), and specific trauma (PCL-S) versions. The PCL shows good temporal stability, internal consistency, test-retest reliability, and convergent validity [47]. PCL-M scores range from 17–85; cut-off scores between 30–35 are indicative of PTSD for those in the general population and for those in Veteran Association Primary Care services cut-off scores between 36–44 are recommended [45]. Depression was assessed by the Patient Health Questionnaire (PHQ-9) [48]. The PHQ-9 has demonstrated reliability, convergent/discriminant validity, and responsiveness to change [49, 50]. PHQ-9 scores range from 0–27; mild, moderate, moderately severe and severe depression are measured with scores of 5–9, 10–14, 15–19 and 20–27 respectively. Anxiety was assessed by the General Anxiety

Disorder (GAD-7) [51]. The GAD-7 has excellent reliability and validity [52, 53]. GAD-7 scores range from 0 to 21; mild, moderate and severe anxiety are measures with scores of 5–9, 10–14 and 15–21 respectively. Perceived stress was assessed by the 10-item Perceived Stress Scale (PSS) [54]. The Perceived Stress Scale exhibits good reliability and convergent validity [55]. PSS scores range from 0 to 40; with low stress categorised as being in the 0–13 range, moderate stress in the 14–26 range, and high perceived stress in the 27–40 range.

## Results and discussion

All participants completed their assigned intervention, with no early departures. At the end of the first of the three sequentially run interventions, participants voluntarily created a Facebook group page to maintain newly established group connections. At the end of subsequent interventions, the participants were made aware of the Facebook group, and encouraged to access it.

**Change in psychological wellbeing.** Participants' mean psychological wellbeing before the intervention and at 2 weeks and 4 months after for each activity type are summarised in Table 3. In order to assess the hypothesised change in psychological wellbeing from before to after the experience, a mixed MANOVA with one between groups factor (intervention type: angling, equine, falconry) and one within-participant factor (3 time points: two weeks pre-intervention, 2 weeks' post intervention and 4 months post-intervention) was conducted on all four measures of psychological wellbeing. It was hypothesised that the analysis would reveal a significant within-participant effect. A Greenhouse-Geisser correction was applied to the within participant degrees of freedom whenever the Mauchly's test of significance was significant, in both experiment one and experiment two.

Results revealed the important significant multivariate within-participant effect reflecting change in psychological measures across time as hypothesized ($F_{(8, 20)} = 4.477$, $p = .003$, partial eta$^2$ = .642). The main effect of intervention group was non-significant ($F_{(8, 50)} = .470$, $p = .872$, partial eta$^2$ = .070), and the interaction of group x time was also non-significant

**Table 3. Experiment one: Summary of measures by activity group and measurement timepoint: Pairwise t-values, Cohen's d and 95% confidence intervals for change in wellbeing over time.**

| Intervention Activity | Measure | M/SD 2 Weeks Prior | M/SD 2 Weeks Post | M/SD 4 Months Post | Pairwise t Prior/2wk post | Pairwise t Prior/4mth post |
|---|---|---|---|---|---|---|
| | | | | | t (d [95% CI]) | t (d [95% CI]) |
| **Angling** | PTSD | 42.36/14.05 | 25.64/9.32 | 28.09/11.22 | 4.27 (-1.126 [-2.03, -0.23]) | 3.2 (- 0.878 [-1.75, -0.003]) |
| n = 11 | Depression | 15.27/6.0 | 9.45/5.30 | 10.82/4.56 | 3.05 (-0.87 [-1.74, 0.005]) | 1.83 (-0.331 [-1.17, 0.51]) |
| | Anxiety | 13.09/5.91 | 10.09/5.24 | 10.64/2.58 | 2.53 (-0.58 [-1.432, -0.27]) | 1.33 (-0.316 [-1.16, 0.53]) |
| | Stress | 25.45/6.76 | 16.36/8.52 | 18.91/5.66 | 2.9 (-0.997 [-1.8, -0.11]) | 2.49 (-0.582 [-1.58, -0.14]) |
| **Equine** | PTSD | 42.38/15.61 | 32.25/14.28 | 36.13/17.02 | 1.75 (-0.595 [-1.60, 0.41]) | 1.19 (-0.44 [-1.43, 0.55]) |
| n = 8 | Depression | 16.63/5.88 | 12.88/8.68 | 14.0/6.63 | 2.33 (-1.265 [-2.34, -0.19]) | 3.28 (-1.313 [-2.40, -0.23]) |
| | Anxiety | 14.00/5.83 | 9.63/7.50 | 11.63/5.73 | 2.43 (-1.03 [-2.07, - 0.01]) | 1.90 (-0.663 [-1.67, 0.34]) |
| | Stress | 25.63/8.18 | 19.13/9.66 | 20.63/8.85 | 3.92 (-1.59 [-2.71, -0.47]) | 3.99 (-1.491 [-2.60, -0.38]) |
| **Falconry** | PTSD | 41.36/9.65 | 28.09/12.75 | 28.73/11.84 | 4.10 (-1.485 [-2.43, -0.51]) | 4.09 (-1.395 [-2.33, -0.46]) |
| n = 11 | Depression | 14.91/5.72 | 10.09/5.3 | 10.00/4.73 | 3.95 (-1.153 [-2.06, -0.25]) | 3.63 (-1.022 [-1.91, -0.13]) |
| | Anxiety | 12.27/3.64 | 8.91/4.46 | 9.00/3.87 | 2.06 (-0.695 [-1.56, 0.17]) | 2.27 (-0.707 [-1.57, 0.16]) |
| | Stress | 22.82/7.04 | 15.45/9.5 | 16.45/9.08 | 4.02 (-1.544 [-2.50, -0.59]) | 3.94 (-1.46 [-2.41, -0.52]) |
| **TOTAL** | PTSD | 42.00/12.62 | 28.30/11.93 | 30.47/13.18 | 5.76 (-1.026[-1.56, -0.49]) | 4.76 (-0.888 [-1.42, -0.36]) |
| N = 30 | Depression | 15.50/5.70 | 10.60/6.28 | 11.37/5.32 | 5.35 (-1.033 [-1.57, -0.50]) | 4.03 (-0.71 [-1.23, -0.19]) |
| | Anxiety | 13.03/5.03 | 9.17/5.75 | 10.30/4.07 | 4.16 (-0.818 [-1.35, -0.29]) | 3.06 (-0.512 [-1.03, 0.002]) |
| | Stress | 24.53/7.12 | 16.77/9.00 | 18.53/7.87 | 5.67 (-1.199 [-1.75, - 0.65]) | 5.33 (-1.03 [-1.57, -0.49]) |

($F$ (16, 42) = .420, $p$ = .969, partial eta$^2$ = .138) indicating that change across time did not vary significantly by activity type. Inspection of the within participant univariate F values confirmed significant reduction across time on all four measures. A Greenhouse-Geisser correction was applied to the degrees of freedom for PTSD and stress measures: ($F$ (1.481, 39.997) = 22.057, $p$ < .001, partial eta$^2$ = .450 (PTSD symptoms)); ($F$ (2, 54) = 15.356, $p$ < .001, partial eta$^2$ = .363 (depression)); ($F$ (2,54) = 10.170, $p$ < .001, partial eta$^2$ = .274 (anxiety); $F$ (1.417, 38.248) = 24.365, $p$ < .001, partial eta$^2$ = .474 (stress)). Means, standard deviations, pairwise $t$-values for the comparison of pre-intervention measures with two week and four month follow-up and Cohen's $d$ (1992) effect sizes and confidence intervals for each intervention are summarised in Table 3. Effect sizes were calculated according to Lenhard & Lenhard [56]. Significant reduction in PTSD symptoms was observed at two-week follow up that was largely sustained at 4 months. Effect sizes were medium to large. A post hoc sensitivity analysis conducted with Gpower 3.9.2 with alpha = 0.05 and power = .80 for a within-participant mean comparison indicated that a sample of 30 participants was sufficient to detect a medium effect (dz = 0.46).

**Clinically significant and reliable change in PTSD symptoms.** While statistical significance provides one index of change, it is also worthwhile to consider if the changes in PTSD symptoms observed might be considered clinically significant or reliable. For Clinically Significant Change (CSC) to be achieved, the level of functioning subsequent to the intervention should fall outside the range of the dysfunctional population, where range is seen as extending to two standard deviations beyond (in the direction of functionality) the mean of the population. The Reliable Change Index (RCI) is calculated using the change in a client's score divided by the standard error of the difference for the measure(s) being used. The Cronbach's alpha used within this calculation was 0.94, which was taken from Sutker, Davis, Uddo and Ditta [57]. Jacobson and Truax [58] provide criteria by which both reliable and clinically significant change can be identified. Where normative data are available for both clinical and non-clinical populations it is advisable to use this criterion (Criterion C) to calculate CSC. For the PCL-M, clinical (mean 63.6, standard deviation 14.1) and non-clinical norms (mean 34.4, standard deviation 14.1) were taken from Weathers et al. [46]. Data for all participants were included in two analyses: baseline-2 weeks post and baseline-4 month follow-up. The pre and post intervention PTSD scores for all 30 participants are illustrated in Fig 2.

When all 30 participants were included in the baseline-2 weeks post analysis 18 (60%) made reliable improvement; 11 (37%) made no reliable change; and only 1 (.03%) deteriorated. In terms of clinical significance 17 (57%) made a clinically significant change in relation to PTSD symptoms. Further to the analysis of the whole sample, an additional analysis of clinically significant change was conducted in which participants who at baseline scored below the clinically significant cut-off score of 41.80 (calculated using Criterion C) on the PCL-M were excluded from the analysis on the basis that if they were not scoring in the dysfunctional range for PTSD diagnosis at baseline it was not possible to make a clinically significant change post-intervention (i.e. move from the clinical range into the non-clinical range). At baseline, 12 participants scored below the clinical cut-off. Of the remaining 18 participants, 14 made clinically significant changes. Fig 2 illustrates the reliable and clinically significant change in PTSD observed.

For the baseline- 4-month follow-up analysis, when all 30 participants were included in the calculation of reliable change 18 (60%) made reliable improvement; 11 (37%) made no reliable change; and only 1 (.03%) deteriorated. For CSC, 16 (53%) achieved clinically significant change in their PTSD symptom score. After excluding those that scored below the clinical cut-off at baseline, 13 met the criteria for making clinically significant change in PTSD symptom scores.

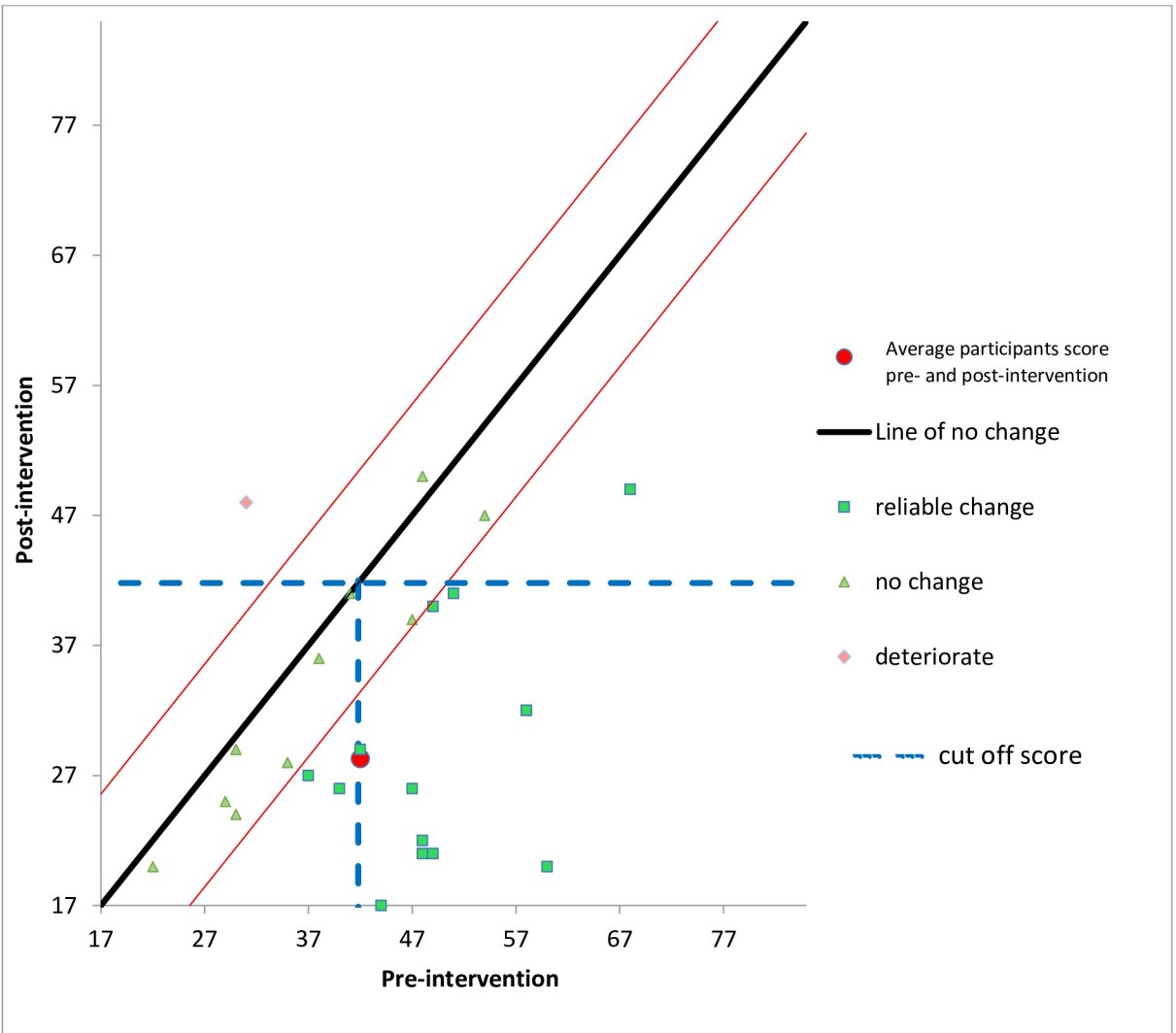

**Fig 2. Reliable and clinically significant change in PTSD symptoms (using Criterion C for PCL-M) at 2 weeks follow up, study one.**
(N = 30).

In sum, findings from experiment one provide evidence that brief outdoor recreational experiences for groups of military veterans improve psychological symptoms. Moreover, our employment of a 4 month follow-up shows that effects were sustained over time. No significant interaction of group by time was obtained between the three groups in the multivariate analysis, indicating that the impact of the experience was not a function of the particular activity undertaken. Nonetheless, the experiment lacked an experimental control group. We therefore conducted a second experiment utilising a waitlist control group. The second experiment compared the effect of an angling intervention with a waitlist control group. Angling was chosen as it is an appealing pastime for our target participants and is a readily accessible and relatively inexpensive hobby to pursue should participants wish to continue after their intervention.

## Experiment two

### Method

**Design and sample size calculation.** Experiment 2 was a waitlist controlled randomized experiment with two independent groups. Ethical considerations meant that it was not possible to have the waitlist group 'wait' until a four-month follow-up was completed for the intervention group (a wait of 18 weeks after recruitment). The controlled comparison of the angling intervention group and the waitlist group therefore is restricted to the two-week follow-up period. Results from experiment one were used to inform the sample size calculation. In the experiment one angling group, the effect of the intervention over time was large when we compared PTSD symptoms before and two weeks after the intervention (d = -1.126, 95% CI [-2.03, -0.23], Table 3), (correlation between within-subject measures, r = .44). Based on the effect of the angling intervention on the average of self-report PTSD symptoms, we computed the required sample size to replicate the effect in a between-participants design where we would compare participants' PTSD symptoms in the intervention group and a control (wait list) group. The a priori required sample size (alpha = 0.05 and power = .80) to detect an effect similar to that found in Experiment 1 was 22, 11 participants per condition.

**Participants, recruitment and randomisation procedure.** Participants were recruited via two military veteran welfare support groups. The researcher attended group meetings to introduce the study and distribute a letter of invitation to participate that included study contact details. Inclusion criteria were to be a military veteran with a formal diagnosis of PTSD by a National Health Service or Ministry of Defence psychiatrist. Participants currently in receipt of psychological therapy were excluded. A total of 57 veterans contacted the study team and were assessed for eligibility by telephone. Of these, 9 did not meet inclusion criteria, and 23 declined to participate after receiving further information (15 did not like the idea of fishing, 6 were unavailable on study dates and 2 had childcare commitments), leaving a total sample size of 25. The 25 participants were randomly allocated using block randomisation to either the angling intervention group (n = 13) or waitlist control group (n = 12). Seven people subsequently did not turn up to the intervention. Recruitment began in July 2015 and data collection continued until December 2015 (4-month follow-up).

**Intervention description.** The intervention involved an angling experience exactly as described in experiment one. In addition to the angling coaches and a mental health practitioner, 3 military veteran participants from experiment one also attended in the role of 'mentor'.

**Measures.** Repeated measures were taken for both groups 2 weeks prior to and 2 weeks following the intervention for the angling group. The waitlist participants were also reassessed two weeks after they subsequently completed the intervention and both groups were followed up 4 months after their own intervention (Fig 3). PTSD symptoms were assessed by the PCL-5 [59]. The PCL-5 is a twenty item self-report measure that assesses the twenty DSM-5 [1] symptoms of PTSD. The PCL-5 is reported to have sound psychometric properties with good internal consistency (.95), test–retest reliability (r .84), and convergent validity with the PCL-S (r .87) [60]. The range for the PCL-5 is 0–80; a cut-off score of 31–33 is suggestive of PTSD. Depression, anxiety and stress were assessed as in the first experiment. Two additional measures were included in experiment two. The Work and Social Adjustment scale [WSAS; 61] is a 5-item measure of impairment in general social functioning and has high internal reliability and sensitivity to treatment effects [62]. The maximum score is 40, with higher scores indicating greater impairment. A final measure was included to assess positive change in psychological growth, as opposed to merely absence of symptoms. The Psychological Wellbeing Post-Traumatic Changes Questionnaire [PWB-PTCQ; 63] has high internal consistency and

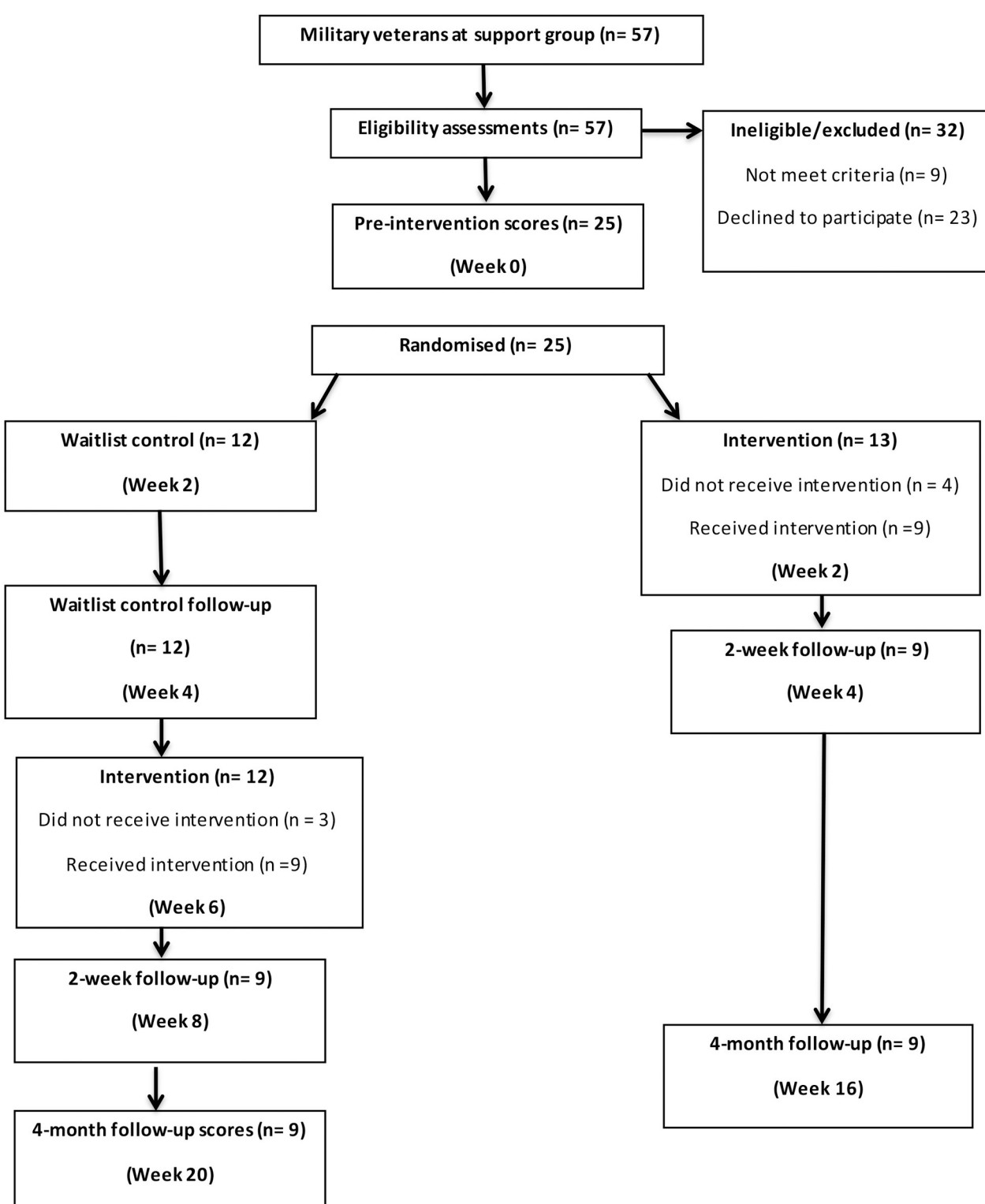

**Fig 3. Summary of measurement and intervention timepoints for intervention and waitlist control group.**

sensitivity to change. Higher scores indicate more positive, post-traumatic change. The maximum score is 90, with scores over 54 indicating the presence of positive change. The measures were taken by an experimenter blind to experimental condition.

## Results and discussion

**Randomisation checks.** Characteristics of participants in the intervention and control groups are shown in the right-hand portion of Table 1. The two groups did not differ in age ($p = .696$), age when joined forces ($p = .625$), years served ($p = .581$) or years since discharge ($p = .517$). All but one of the participants were men, and equal numbers in each condition were unemployed and taking psychotropic medication. A between groups MANOVA showed that the two groups did not differ at baseline across psychological measures ($F (6, 11) = 1.319$, $p = .326$). The Univariate $F$ values for the comparison of psychological measures at baseline are displayed in Table 4. These analyses confirm that randomisation was successful in distributing key individual differences evenly between groups.

**Comparison of control and intervention groups.** The key analysis relates to the comparison of pre- (week 0) and post-intervention measures for the intervention and control groups taken at week 4 (see Fig 3). A mixed MANOVA with one between-groups factor (intervention vs. waitlist control) and repeated measures on all outcomes (baseline vs. 2 weeks post intervention) revealed a non-significant main effect of group ($F (6, 11) = 2.061$, $p = .141$, partial eta$^2 = .529$), a significant main effect of time ($F (6,11) = 3.626$, $p = .031$, partial eta$^2 = .664$) that was qualified by the important interaction of group by time ($F (6,11) = 3.547$, $p = .033$, partial eta$^2 = .659$). Inspection of the univariate $F$ values showed that all variables contributed to this interaction effect ($F (1,16) = 8.445$, $p = .01$, partial eta$^2 = .345$ (PTSD symptoms), $19.982$, $p < .001$, partial eta$^2 = .555$ (depression), $11.225$, $p = .004$, partial eta$^2 = .412$ (anxiety), $13.474$, $p = .002$, partial eta$^2 = .457$ (work and social adjustment), $4.610$, $p = .047$, partial eta$^2 = .224$ (stress), $12.148$, $p = .003$, partial eta$^2 = .432$ (post-traumatic growth)). A summary of the means, standard deviations and univariate between groups $F$ tests at time two are summarised in Table 4. At two weeks post-intervention, participants in the intervention group had

**Table 4. Summary of measures at baseline (2 weeks prior) and 4 weeks (2 weeks post-intervention) by intervention and waitlist control groups and effect size of mean difference between intervention and control groups at 4 weeks.**

| Measure | Timepoint | Intervention Group | Waitlist Control Group | Between Groups Univariate F value df (1,16) | Cohen's d [95% CI] |
|---|---|---|---|---|---|
| | | *(n = 9)* | *(n = 9)* | | |
| | | *Mean (SD)* | *Mean (SD)* | **F (p)** | |
| PCL (PTSD) | Baseline | 47.00(15.98) | 55.67(12.35) | 1.657 (.216) | -1.197 [-2.2, -0.194] |
| | 4 weeks | 33.78(18.45) | 53.67(14.56) | 6.503 (.021) | |
| PHQ (Depression) | Baseline | 18.00(6.16) | 19.00(4.80) | .148 (.706) | -1.66 [-2.732, -0.589] |
| | 4 weeks | 10.44(6.20) | 19.22(4.18) | 12.389 (<0.001) | |
| GAD (Anxiety) | Baseline | 13.44 (4.00) | 15.89(3.79) | 1.770 (.202) | -1.623 [-2.688, -0.558] |
| | 4 weeks | 8.22(4.76) | 15.78 (4.55) | 11.841 (.003) | |
| PSS (Stress) | Baseline | 24.11(7.17) | 27.56 (4.75) | 1.445 (.247) | -1.988 [-2.956, -0.747] |
| | 4 weeks | 17.56 (5.81) | 29.11(6.64) | 15.429 (.001) | |
| WSAS (Work and social adjustment) | Baseline | 21.89 (7.04) | 23.22(10.89) | .095 (.762) | -0.787 [-1.746–0.172] |
| | 4 weeks | 15.44 (8.35) | 22.556 (9.671) | 2.787 (.114) | |
| PWB-PTCQ Posttraumatic wellbeing | Baseline | 47.56(11.09) | 49.50(16.33) | .082 (.778) | 1.279 [0.265–2.293] |
| | 4 weeks | 65.78(8.98) | 48.89(16.37) | 7.360 (.015) | |

significantly lower PTSD symptomology, depression, anxiety, and perceived stress relative to controls, and also reported post-traumatic growth relative to controls.

**Change in wellbeing across time.** Since all participants eventually completed the angling activity experience it was also possible to examine change across time for all 18 participants combined. These analyses provide a replication and extension of findings from experiment one. A MANOVA with repeated measures was conducted across all measures at 3 timepoints (2 weeks prior to intervention, 2 weeks post and 4 months post; see Fig 3 for respective weeks of measurement). Results revealed a significant multivariate within-participants effect ($F$ (12, 6) = 7.492, $p$ = .011, partial eta$^2$ = .937). Inspection of univariate within participant effects showed that change in psychological measures across 3 time points was significant for all measures ($F$ (2,34) = 26.254, $p < 0.0001$, partial eta$^2$ = .607 (PTSD symptoms), 43.799, $p < 0.0001$, partial eta$^2$ = .720 (depression), 25.730, $p < 0.0001$, partial eta$^2$ = .602 (anxiety), 16.123, $p < 0.0001$, partial eta$^2$ = .487 (perceived stress), 13.493, $p < 0.0001$, partial eta$^2$ = .442 (work and social adjustment), 15.404, $p < 0.0001$, partial eta$^2$ = .475 (posttraumatic growth)). A summary of means, standard deviations, pairwise t-tests comparing pre-intervention with 2-week and 4-month follow-up respectively is displayed in Table 5. Results show that participants experienced a decline in PTSD symptoms, depression, anxiety and stress, with improved work and social functioning and increased post-traumatic growth. Cohen's d effect sizes were calculated according to Lenhard & Lenhard [56]. As in experiment one, effect sizes were large and sustained to 4 months.

**Clinically significant and reliable change in PTSD symptoms.** As in experiment one, further analyses were conducted to evaluate clinical significance of findings both between the baseline and 2-week post intervention and the baseline and 4-month follow-up time points. As previously, analyses were conducted for the PCL-5 symptoms first including all participants and then including only those who scored above the clinical cut-off—score of 33 as recommended in Wortmann et al., [60] at pre-intervention baseline. In computing the Reliable Change Index, the Cronbach's alpha used within this calculation was 0.95, derived from Wortmann et al., [60].

For the whole sample pre-post intervention, 12 (67%) made reliable improvement; 6 (33%) did not reliably change and no-one deteriorated in their PTSD symptoms. In relation to CSC, 5 (28%) made clinically significant change in their PTSD symptom scores. When those who scored below the clinical cut-off at baseline were excluded ($n$ = 3), 5 made clinically significant change in their PTSD symptom scores. Fig 4 illustrates the reliable and clinically significant changes pre-post intervention for the 18 participants.

For the whole sample baseline-4 month follow-up, the same pattern of results were found, 12 (67%) made reliable improvement; 6 (33%) did not reliably change and no participant reported deterioration in their PTSD symptoms. In relation to CSC, 5 (28%) made clinically significant change in their PTSD symptom scores. When those who scored below the clinical cut-off at baseline were excluded (n = 3), 5 made clinically significant change in their PTSD symptom scores.

## General discussion

Research on the therapeutic potential of outdoor recreational experiences for military veterans with PTSD is in its infancy. The goal of the present studies was to provide formal evaluation of the potential impact of brief outdoor activity experiences amongst military veterans with diagnosed PTSD who were not receiving any form of psychological therapy. The results of the two experiments, comprising locally delivered, outdoor recreational interventions amongst veteran peers, demonstrate not only the potential of motivating veterans with PTSD to engage with

**Table 5. Summary of measures at two weeks pre-intervention, two weeks post-intervention and 4 months post-intervention for combined sample (Experiment 2: N = 18): Pairwise t value, Cohen's d and 95% confidence interval.**

| Measure | Timepoint | Mean (SD) | Pairwise t value, df = 17 (p-value) comparison with baseline pre intervention | Cohen's d [95% CI] For comparison with baseline pre-intervention |
|---|---|---|---|---|
| PTSD symptoms (PCL-5) | Pre-intervention | 50.33 (15.15) | | |
| | 2 weeks post | 34.56 (15.52) | 5.399 (< .001) | -1.287 [-2.01, -0.57] |
| | 4 months post | 37.06 (15.41) | 5.346 (< .001) | -1.27 [-1.99, -0.55] |
| Depression (PHQ-9) | Pre-intervention | 18.61 (5.150) | | |
| | 2 weeks post | 10.89 (5.54) | 7.830 (< .001) | -1.922 [-2.71, -1.13] |
| | 4 months post | 12.50 (5.04) | 6.113 (< .001) | -1.43 [-2.16, -0.7] |
| Anxiety (GAD-7) | Pre-intervention | 14.61 (4.33) | | |
| | 2 weeks post | 9.06 (4.70) | 5.644 (< .001) | -1.389 [-2.12, -0.66] |
| | 4 months post | 10.17 (3.99) | 4.710 (< .001) | -1.068 [-1.77, -0.37] |
| Stress (PSS) | Pre-intervention | 26.61 (7.18) | | |
| | 2 weeks post | 18.94 (5.82) | 4.302 (< .001) | -0.928 [-1.62, -0.24] |
| | 4 months post | 19.78 (5.91) | 3.779 (< .001) | -0.819 [-1.50, -0.14] |
| Work and social adjustment (WSAS) | Pre-intervention | 22.22 (8.21) | | |
| | 2 weeks post | 15.33 (9.16) | 4.040 (< .001) | -1.015 [-1.71, -0.32] |
| | 4 months post | 16.89 (8.65) | 3.319 (< .004) | -.804 [-1.48, -0.13] |
| Post traumatic growth (PWB-PTCQ) | Pre-intervention | 48.22 (13.58) | | |
| | 2 weeks post | 63.78 (11.01) | -5.549 (< .001) | 1.206 [0.5, -1.92] |
| | 4 months post | 62.11 (14.35) | -4.326 (< .001) | 1.051 [0.35–1.75] |

such an approach but also its potential clinical usefulness. In both studies and in line with our hypotheses, veterans showed a clear and sustained improvement in symptomology relating to PTSD, depression, anxiety and stress as a consequence of the experience. In addition, in the second experiment, measures of post-traumatic growth alongside work and social adjustment were both found to improve following the intervention. Experiment two also provided important evidence that participants randomly allocated to the therapeutic recreation experience experienced symptomatic change that differed significantly from those randomised to a wait-list control group.

These findings not only add to the previous literature in terms of the potential usefulness of outdoor recreational interventions in the treatment of PTSD and its comorbidities but also extend and strengthen the evidence in three essential ways: Firstly, only participants who were not currently in receipt of psychotherapy were enrolled onto the studies; this is important as some previous studies have included participants in receipt of therapy, making it difficult to

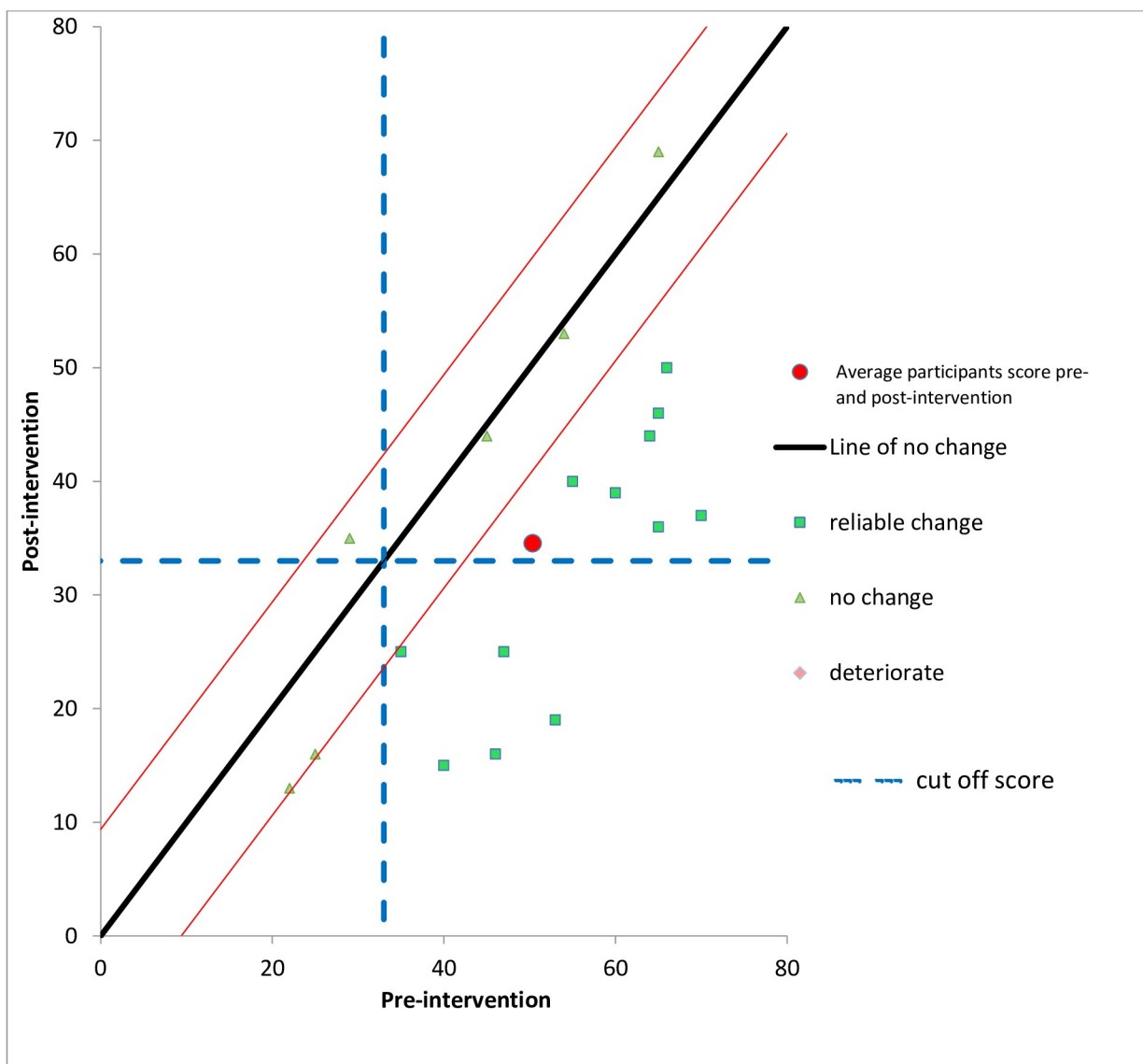

**Fig 4. Reliable and clinically significant change in PTSD symptoms (using external criterion for PCL-5) at two weeks' follow-up, Experiment two.** (N = 18).

distinguish benefit derived from the intervention from that derived from ongoing psychotherapy; Secondly, the interventions employed in the present studies were all relatively brief, conducted close to participants' homes and at low cost, demonstrating not only the potential but also the relative ease and affordability of such an approach; Thirdly, the second experiment employed a control (waitlist) condition, thereby facilitating a more robust interpretation of the findings—something that has only been done twice previously in the literature of this field [35].

Another aspect of the experiments reported here is that all participants in both studies had a National Health Service or Military Physician diagnosis of PTSD. They were followed up to 4 months and data subjected to additional analyses of reliable and clinically significant change. Previous studies of an outdoor experience directed at military veterans have not considered reliable or clinically significant change, relying upon statistical significance that assesses mean

difference without reference to the relevance of the effect [35]. For example, statistical significance may be obtained if a participant score changes within a non-clinical range. Analyses from both studies showed that a substantial proportion of participants showed reliable improvement in PTSD symptoms 4 months following the intervention, figures that are in line with those reported for psychotherapy [e.g. 21].

A variety of different experiences were explored in the present studies. They all incorporated common elements of our procedure, namely exclusive use of outdoor facilities by veterans with diagnosed PTSD, safety briefing, allocation to professional recreation trainers in a ratio of 2:1, communal food preparation and socialising by participating veterans. No formal psychological therapy was offered or provided to participants during the experience. Findings of the experiment one multivariate analysis showed that within participants' change in wellbeing across time was not qualified by type of activity pursued. This is the first study to compare different outdoor experiences and findings suggest that the type of outdoor activity experienced was not necessarily a critical factor in the results obtained. Active intervention elements appear to be those held in common, namely professional instruction in an outdoor recreational activity in the company of peers [c.f. 36]. Additionally, evidence suggests that readily available local environments such as a stocked fishing lake can deliver benefits similar to those previously obtained in more wild and exotic locations [35].

## Strengths and limitations

The present studies have many strengths. They studied military veterans with diagnoses of PTSD from a national health service or military psychiatrist, and employed a formal procedure for the outdoor activity experience. Measures were taken at baseline and participants were followed for four months. No participant was lost to follow-up. Most important, experiment two employed a waitlist control group and random allocation to condition.

Nonetheless, a number of limitations should be acknowledged. Due to ethical considerations, the comparison of the waitlist control and experimental groups was restricted to two weeks and it would be desirable to replicate this effect over a longer time period. Although waitlist participants received the same phone calls and administration of measures, future studies might also employ an active rather than passive waitlist control condition. This is important as it would help to address any possible issues of participants' expectations from the experience and respondent bias (e.g. acquiescence and social desirability bias). It should be acknowledged that the majority of participants were men and generalisation to women veterans with PTSD is therefore uncertain. The intervention format, together with the hard to reach population necessitates the involvement of small groups of veterans, as previously noted by Bird [31]. Nonetheless the current evidence presented shows that these small sample sizes were sufficiently powered to detect the effect sizes observed. A total of 48 veterans with untreated PTSD, mostly with chronic PTSD, took part in these studies and evidence showed that more than half of the participants experienced change in PTSD symptoms that is indicative of reliable change of clinical significance. Some participants in these studies have subsequently made social and economic gains by entering education, employment, deciding to engage in psychotherapy, and regaining access to children. Some have trained as fishing coaches. All now have access to a support group of veterans with PTSD with shared experience of learning to adapt to civilian life.

## Conclusions

Accumulating evidence across nations provides support for the role of therapeutic outdoor recreation in addressing the particular psychological needs of military veterans with PTSD.

The use of outdoor recreational activities to ameliorate PTSD and co-morbid symptoms and improve well-being in military veterans appears to be a viable and useful treatment option. Further research is indicated to evaluate and promote the availability of this and other programs to the millions of men and women adapting to life after war. This is particularly important for a clinical population who are traditionally thought to be difficult to engage in formal therapies and yet whose continued ill health is a considerable burden to the individuals, families and communities involved.

## Supporting information

**S1 File. Protocol—Pilot study.**
(DOCX)

**S2 File. Protocol—RCT.**
(DOCX)

**S1 Checklist. CONSORT 2010 checklist of information to include when reporting a randomised trial**∗**.**
(DOC)

## Acknowledgments

We would like to thank Cliff Davis and Crafty Catcher (for supplying bait and angling coaches), Danny Fairbrass at Korda International (for supplying tackle and bait), Les Webber MBE (for access to lakes), Lavenham Falconry and the Garrison Stables (Colchester).

## Author Contributions

**Conceptualization:** Mark Wheeler, Nicholas R. Cooper, Jamie Hacker Hughes, Tim Rakow.

**Data curation:** Mark Wheeler, Leanne Andrews, Sheina Orbell.

**Formal analysis:** Mark Wheeler, Leanne Andrews, Marie Juanchich, Tim Rakow, Sheina Orbell.

**Investigation:** Mark Wheeler, Nicholas R. Cooper.

**Methodology:** Mark Wheeler, Nicholas R. Cooper, Leanne Andrews, Marie Juanchich, Tim Rakow, Sheina Orbell.

**Project administration:** Mark Wheeler, Nicholas R. Cooper.

**Supervision:** Nicholas R. Cooper, Tim Rakow, Sheina Orbell.

**Validation:** Nicholas R. Cooper.

**Writing – original draft:** Mark Wheeler, Nicholas R. Cooper, Sheina Orbell.

**Writing – review & editing:** Mark Wheeler, Nicholas R. Cooper, Leanne Andrews, Jamie Hacker Hughes, Marie Juanchich, Tim Rakow, Sheina Orbell.

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
