## [Decision Letter · Decision Letter 0]

18 May 2020

PONE-D-19-35181

Outdoor recreational activity experiences improve psychological wellbeing of military veterans with post-traumatic stress disorder: positive findings from a pilot study and a randomised controlled trial

PLOS ONE

Dear Dr. Cooper,

Thank you for submitting your manuscript to PLOS ONE. After careful consideration, we feel that it has merit but does not fully meet PLOS ONE’s publication criteria as it currently stands. Therefore, we invite you to submit a revised version of the manuscript that addresses the points raised during the review process.

The manuscript has been evaluated by two reviewers, and their comments are available below.

The reviewers have raised a number of concerns that need attention. They request additional information on methodological aspects of the study and the analyses, and they have raised some concerns about the conclusions and their interpretation. 

Could you please revise the manuscript to carefully address the concerns raised?

We would appreciate receiving your revised manuscript by Jun 29 2020 11:59PM. To enhance the reproducibility of your results, we recommend that if applicable you deposit your laboratory protocols in protocols.io, where a protocol can be assigned its own identifier (DOI) such that it can be cited independently in the future. For instructions see: http://journals.plos.org/plosone/s/submission-guidelines#loc-laboratory-protocols

We look forward to receiving your revised manuscript.

Kind regards,

Vanessa Carels

Staff Editor

PLOS ONE

Journal Requirements:

"I have read the journal's policy and the authors of this manuscript have the following

competing interests: Following the results of the studies reported in this manuscript

(and subsequent work), two authors (Wheeler & Cooper) have set up a Community

Interest Company (iCARP CIC) in order to provide more outdoor pursuit experiences

for more veterans. They receive no income for this venture."

Reviewers' comments:

Reviewer's Responses to Questions

**Comments to the Author**

1. Is the manuscript technically sound, and do the data support the conclusions?

Reviewer #1: Yes

Reviewer #2: Partly

2. Has the statistical analysis been performed appropriately and rigorously?

Reviewer #1: Yes

Reviewer #2: No

3. Have the authors made all data underlying the findings in their manuscript fully available?

Reviewer #1: Yes

Reviewer #2: Yes

4. Is the manuscript presented in an intelligible fashion and written in standard English?

Reviewer #1: Yes

Reviewer #2: Yes

5. Review Comments to the Author

Reviewer #1: The two related studies reported in the manuscript investigate the short- and mid-term psychological effects of very brief (1-day) recreational interventions for participants with PTSD. Although considerable favourable changes in several meaures are reported, these results are, in my opinion, of doubtful value by themselves fort he following reasons. Firstly, the outcome measures are subjective, self-reported ones obtained in participants who were not blinded tot he intervention. Thus, positive effects may well occur due to participants‘ expectations or to general beneficial effects due to receiving attention. The comparison with the waiting list in experiment two unfortunately does not deal effectively with this deficit. Secondly, although this is entirely a layman’s opinion, I cannot imagine that a single day’s activity could yield a meaningful, enduring benefit in people who have been suffering PTSD for many years.

Further important weaknesses:

(1) the calculation oft he reliable change index (RCI) is incompletely described. The calculation of ‚standard error‘ must incorporate a reliability measure such as Cronbach’s alpha in ordert o arrive at a measure of measurement error rather than variability.

(2) The exclusion of cases with non-dysfunctional values at baseline in Figs. 2 and 4 induces an element of ‚regression tot he mean‘ in the subsequent changes and is thus biased in the direction of favourable changes.

Minor points (by line number)

38-39 Ambiguity in AND and OR

233 Please state the randomisation method

Tab. 1 years military service and years since leaving are mean (SD)?

244 Please state explicitly: only one session on one day

294-5 What is meant by ‚first‘ and ‚subsequent‘ interventions?

305 Main effect of intervention group is not interpretable, since all groups were equally treated at baseline

316 Describe or cite method for effect sizes, rather than just citing the web-site

346 ‚Clinical range‘ means dysfunctional range?

372 Why was angling chosen for experiment 2?

377-8 Which ethical considerations preclude a 4-month wait (they have already ‚waited‘ so many years!)?

388 ‚Effect‘ means between-groups at post-intervention timepoint?

409 Make clear that ‚intervention‘ refers to intervention period of the intervention group, not the subsequent intervention after waiting

Tab. 3 Why refer to ‚4 weeks‘ – better as in Table 2 ‚2 weeks prior‘ and ‚2 weeks post‘

521 There were 3, not 4, interventions?

536 Why would including participants with psychotherapy bias the results?

573 Active control group is important in order to reduce ‚expectation‘ or ‚attention‘ bias

Reviewer #2: Dear Editor,

Thank you for inviting me to review this manuscript. I believe that the topic is very interesting and that the authors put effort in drafting this work. The aim of this study was to enhance the knowledge regarding brief outdoor activity experiences amongst military veterans with diagnosed PTSD. After reading the manuscript, I was also left with a few questions about the work and I believe addressing these will enhance the contribution to the literature. Questions/concerns are outlined by manuscript section below.

Introduction

It is not clear what are each study objectives and hypothesis

1. Study objectives: Abstract [line36]: “Two studies were conducted to evaluate this possibility”. The authors should clearly state what where the specific objectives for each study in the introduction, and in the abstract.

2. Study hypothesis [line 201-5]: “It was hypothesized that participants would experience a reduction in symptomology as a consequence of the experience.” …“The hypothesis that experimental participants would experience benefit relative to controls was supported.”

• The study hypothesis are vague and not clear. For example what do the authors refer in “symptomology “or “benefit”?

• To improve it would recommended stating for each study hypotheses what are the outcome measures and detraction of association.

Methods:

I recommend strengthening and clarifying the methods section, as detailed below.

1. Experiment One Design [line 219]: Could the authors please clarify, based on my understanding all 3 group received intervention, there is no control group as such, this design isn’t RCT.

2. Random allocation: There could be much more clarity in the Methods about who and what methods where used to generate random allocation to interventions groups.

3. Participants: “Eligibility criteria: military veteran with a formal diagnosis of PTSD by a National Health Service or Ministry of Defence psychiatrist. None were currently receiving psychological therapy for PTSD.”

• It isn’t clear what methods and process where used to verify formal diagnosis of PTSD [medical recorders, was it based on self –report?]

• Given that ‘military veteran’ differs between countries and governments, could the authors please clearly how they define ‘veteran’ in the study?

• There could be much more clarity in the Methods regarding the screen process/data collection, the authors present in Table 1. The summary of participant characteristics [include years’ military service, years since leaving service etc. ]. However, it is not clear how the data was collect. This should have been clearly signposted in the Methods.

4. Intervention descriptions: There could be much more clarity in the Methods about interventions, seating it accrued in each study. Perhaps with the addition of a table which illustrates each intervention group in in each study?

• What was the duration to each session in each intervention group (angling, horse husbandry and riding, falconry and archery)

• What was the activities in each intervention groups? what tools been used how long ? etc..

• How many professional coaches in each intervention group

• What was the training, that professional coaches who provide the intervention had to undertake?

Etc…

5. Measurements:

• In the aims [line 193] the authors stated: We designed two experiments to contribute to and extend previous literature by providing an evaluation of the feasibility and effects of brief peer group outdoor recreational”. Could the authors please clarify, how feasibility of the intervention was assessed – please describe how it was done and be how?

• To help the reder interpret the effect size, please provide the score range for each of the outcome measure that been used in the study.

6. Statistical Analyses: Some additional detail and clarification of the data analysis steps used would be helpful. Given the small sample could the authors please add additional information which analysis were used to evaluated inferential analyses and relevant statistical assumptions (including normality, linearity, homoscedasticity).

 

7. The following statement are not fully supported by the results of this study: “The results of the two experiments, comprising four local, outdoor recreational interventions, demonstrate not only the feasibility of motivating veterans with PTSD to engage with such an approach but also its potential clinical usefulness”.

8. I am concerned that the authors may be over-interpreting the results of the study. The authors stated in the general discussion: “Another aspect of the experiments reported here is that all participants in both studies had an NHS or Military Physician diagnosis of PTSD. They were followed up to 4 months and data subjected to additional analyses of reliable and clinically significant change. Previous studies of an outdoor experience directed at military veterans have not considered reliable or clinically significant change, relying upon statistical significance that assesses mean difference without reference to the relevance of the effect”.

However:

• It is not clear what was the diagnosis or the severity of PTSD. Potently it could be that the Veteran that agreed to participants were diagnosed with PTSD who is less severe.

• Furthermore, in the analysis it is not clear way the authors used Jacobson and Truax [not in veteran population] nor way they decided to remove segments from the tool. Which led to reduction in the sample to 18.

The PCL-M is among few validated measures of PTSD severity both in line with the DSM-5 and demonstrating excellent psychometric properties. Preliminary cut-scores of both 33 and 38 have been recommended as indicating PTSD presence. These validation efforts were implemented among veteran samples, and optimal PCL cut-scores vary across populations depending on factors like trauma type. It would recommended to use the full valid tool PCL-5 and updated research. Please see: Kazdin AE. Methodological issues and strategies in clinical research. 3rd ed.

6. PLOS authors have the option to publish the peer review history of their article (what does this mean?). If published, this will include your full peer review and any attached files.

Reviewer #1: Yes: Jeremy Franklin

Reviewer #2: No

---

## [Author Response · Author response to Decision Letter 0]

12 Jun 2020

Response to Reviewers

We would like to begin by thanking our editor and both of our reviewers for their careful reading of our paper and for their constructive comments and suggestions. For ease of reading, we have formatted our responses in blue text.

Editor:

The reviewers have raised a number of concerns that need attention. They request additional information on methodological aspects of the study and the analyses, and they have raised some concerns about the conclusions and their interpretation. 

We are grateful to the reviewers for noting aspects of our manuscript that required further elaboration or clarification in order to enhance the contribution of the work to the literature. We have outlined our response to each of the points raised by your reviewers in the following sections. In particular we have added information to clarify details of our methods and analysis. We have also reflected upon our interpretation and discussion and conclusion sections to ensure that the claims are not overstated. We consider that these revisions have resulted in an improved manuscript and look forward to receiving your response. We have also checked the layout and style requirements for publication. 

Reviewer #1: The two related studies reported in the manuscript investigate the short- and mid-term psychological effects of very brief (1-day) recreational interventions for participants with PTSD. 

Although considerable favourable changes in several measures are reported, these results are, in my opinion, of doubtful value by themselves for the following reasons. 

• Firstly, the outcome measures are subjective, self-reported ones obtained in participants who were not blinded to the intervention. Thus, positive effects may well occur due to participants‘ expectations or to general beneficial effects due to receiving attention. The comparison with the waiting list in experiment two unfortunately does not deal effectively with this deficit. 

o Thank you for taking the time to review our paper and for your evaluation of the paper as technically sound, and that the data support the conclusions. As discussed in the introduction (lines 176-216) and noted by reviewer 2 the studies extend existing literature concerning outdoor activity interventions for military veterans. The current studies build upon prior research in several ways, for example by employing power analyses, a control group and longer follow up. The measures we used are well respected, validated and established measures of PTSD, depression, anxiety etc. commonly employed to assess psychotherapy outcomes. (Please also see response to Reviewer 2’s 1st comment below). Nonetheless, we do not assert yet that this is a definitive trial of this approach and in the discussion sections acknowledge the limitations of a waitlist control group, for example, and call for further research, that is indicated by this contribution to the literature. Also, we have added to the discussion (please see response to your final comment below and lines 627-631 with respect to the potential issue of bias and expectancy.

• Secondly, although this is entirely a layman’s opinion, I cannot imagine that a single day’s activity could yield a meaningful, enduring benefit in people who have been suffering PTSD for many years.

o These interventions perhaps need to be viewed in the context of a population who employ avoidant coping responses and have done so for many years. Some participants rarely left their homes. This is explained in the introduction (pages 4-8). The intervention has three important elements: being amongst other veterans with PTSD (peers), learning a new activity and being outdoors (lines 136-165). We agree that the results may surprise people but would point out that these single days comprise many hours (equating to many sessions of psychotherapy if looked at another way), and to our use of established measures of trauma, mental health and wellbeing. Our findings progress the literature in this field and provide new, improved evidence that paves the way for more sophisticated research designs and research funding that may address the limitations you pose. We have fully acknowledged these limitations in the discussion section of the paper. 

o Perhaps also very important, organisations around the world are offering interventions of this kind to this client group (and in some cases at a fee) and our research contributes to development of an evidence assessment for this work, including assessment that it does no harm and development of more detailed protocols. 

Further important weaknesses:

(1) the calculation of the reliable change index (RCI) is incompletely described. The calculation of‚ standard error‘ must incorporate a reliability measure such as Cronbach’s alpha in order to arrive at a measure of measurement error rather than variability.

We have now addressed this issue by adding:

1) “The Cronbach’s alpha used within this calculation was 0.94, which was taken from Sutker, Davis, Uddo and Ditta (57)” to lines 370-1 and

2) “In computing the Reliable Change Index, the Cronbach’s alpha used within this calculation was 0.95, derived from Wortmann et al., (60).” To lines 542-3

(2) The exclusion of cases with non-dysfunctional values at baseline in Figs. 2 and 4 induces an element of ‚regression to the mean‘ in the subsequent changes and is thus biased in the direction of favourable changes.

The Figures 2 and 4 did show the data only for those participants with dysfunctional values at baseline. Since both reviewers prefer a different approach we have now presented new figures 2 and 4 that show data from the entire samples in experiments one and two. These revised figures with full samples now permit the reviewer and reader to view the change in scores of every participant and consider for themselves the changes. We agree that exclusion of those who (conventionally) would not be considered dysfunctional at baseline may cause regression to the mean. On the other hand, not excluding these participants means that calculation of clinically significant change might inadvertently refer to a change within the non clinical (non dysfunctional) range- for example a person might obtain a significantly reduced PTSD symptom score but this change could be within the range of non-dysfunctional values. This issue is debated in the literature. We therefore chose to present the calculations of CSC with and without exclusions and reported this in full in our manuscript for transparency. Edits have been made to the text and figure captions on lines 378-9 & 391-2 (experiment 1) and lines 548-552 (experiment 2) to ensure this is clear.

Minor points (by line number) – please note – line numbers have now changed from the original; we have referred to the new line numbers in blue text.

38-39 Ambiguity in AND and OR

Corrected, line 41 – “angling, equine care, or archery and falconry combined”

233 Please state the randomisation method

The manuscript has been updated to include this information: “Participants were randomly allocated (using an online randomisation tool – www.random.org) to either an angling, equine husbandry or falconry and archery recreational experience.” (now lines 251-2). 

Tab. 1 years military service and years since leaving are mean (SD)?

Corrected – (SD) added in table. Line 2.

244 Please state explicitly: only one session on one day

Yes that is correct. We have now clarified - “Each intervention was designed to deliver a day-long outdoor recreational experience”. Now line 265.

294-5 What is meant by ‚first‘ and ‚subsequent‘ interventions?

As noted on line 236 the three interventions with independent samples (angling, equine and falconry and archery combined) took place sequentially. This was simply relating to the fact that after the first intervention session that we ran, a Facebook page was set up. At subsequent sessions (with different participants), we made the participants aware of the Facebook page. We have re-written line 329 so that it now states: “at the end of the first of the three sequentially run interventions…”.

305 Main effect of intervention group is not interpretable, since all groups were equally treated at baseline

Here, a main effect of group refers to a difference between the different types of intervention (angling v equine v falconry) – there was no effect of group, indicating that all interventions had a somewhat similar effect. Now line 343-6.

316 Describe or cite method for effect sizes, rather than just citing the web-site

Now line 354. The reference to Lenhard & Lenhard (2016) has been added. 

346 ‚Clinical range‘ means dysfunctional range?

We have added “not scoring in the dysfunctional range for PTSD diagnosis” to clarify. Now lines 384-5.

372 Why was angling chosen for experiment 2?

We have added the following for clarification: “Angling was chosen as it is an appealing pastime for our target participants and is a readily accessible and relatively inexpensive hobby to pursue should participants wish to continue after their intervention.” Now lines 408-10. Results of experiment one showed that the 3 activities we explored produced very similar results. 

377-8 Which ethical considerations preclude a 4-month wait (they have already ‚waited‘ so many years!)?

Now line 416. 

There were 2 main reasons. (Perhaps they could be considered both practical and ethical). (1) We were keen to retain participants allocated to the waitlist control group to the experiment and (2) the charity that assisted us in recruitment also felt it unethical to offer and then enforce a 4 month wait. We are happy to change this to “Practical considerations” if you prefer.

388 Effect‘ means between-groups at post-intervention timepoint?

The power calculation was based on the repeated measures effect (2 weeks pre v 2 weeks post) for the angling intervention group. Now lines 421-2.

409 Make clear that ‚intervention‘ refers to intervention period of the intervention group, not the subsequent intervention after waiting

Done. Now lines 449.

Tab. 3 Why refer to ‚4 weeks‘ – better as in Table 2 ‚2 weeks prior‘ and ‚2 weeks post‘

Simply due to space in the table. We have added some clarity in the table legend to help make the comparison with table 2 easier. Lines 505-6. NB Table 3 is now Table 4 and Table 2 is Table 3.

521 There were 3, not 4, interventions?

We were referring to the 3 interventions in the first experiment and the 1 in the second. As this may cause some confusion, we have now removed reference to the number of interventions. Now line 569.

536 Why would including participants with psychotherapy bias the results?

This is a very pertinent question, is something that possibly obscures the findings of previous studies done in this area and was one of the driving factors for us undertaking these two studies. We introduce our rationale in the introduction in lines 188-94: “First, we excluded all potential participants who were in receipt of psychotherapy. Military veterans often avoid psychotherapeutic treatment for many years, and as noted by Gelpkof (32), interventions such as these may be particularly valuable for those not yet ready to engage, who have not benefitted or who have been treatment drop outs. Also, the inclusion of participants who were concurrently receiving psychotherapy in some previous studies (e.g. 40) makes it difficult to discern if benefit arose from the intervention or from ongoing treatment”. We see this as one of the strengths of our study and also refer to it again in the discussion – lines 582-6.

573 Active control group is important in order to reduce ‚expectation‘ or ‚attention‘ bias

This is a fair point and one that is related to your 1st point (above). We have added “This is important as it would help to address any possible issues of participants’ expectations from the experience and respondent bias (e.g. acquiescence and social desirability bias)” to lines 628-30.

Reviewer #2: 

Thank you for inviting me to review this manuscript. I believe that the topic is very interesting and that the authors put effort in drafting this work. The aim of this study was to enhance the knowledge regarding brief outdoor activity experiences amongst military veterans with diagnosed PTSD. After reading the manuscript, I was also left with a few questions about the work and I believe addressing these will enhance the contribution to the literature. Questions/concerns are outlined by manuscript section below.

Thank you for your positive reaction to the manuscript and very useful suggestions to enhance the contribution. We outline our responses below. 

It is not clear what are each study objectives and hypothesis

1. Study objectives: Abstract [line36]: “Two studies were conducted to evaluate this possibility”. The authors should clearly state what where the specific objectives for each study in the introduction, and in the abstract.

Thank you for helping us to make this more explicit. We have now added in a more specific aim in lines 35-39: ‘In particular, these studies aimed to test the hypothesis that a brief group outdoor activity would decrease participants’ symptoms as assessed by established measures of PTSD, depression, anxiety and perceived stress, and increase participants’ sense of general social functioning and psychological growth.’

2. Study hypothesis [line 201-5]: “It was hypothesized that participants would experience a reduction in symptomology as a consequence of the experience.” …“The hypothesis that experimental participants would experience benefit relative to controls was supported.”

• The study hypothesis are vague and not clear. For example what do the authors refer in “symptomology “or “benefit”?

• To improve it would recommended stating for each study hypotheses what are the outcome measures and detraction of association.

Again, thank you for helping us to make this more explicit. We have now added: ‘It was hypothesized that participants would experience a decrease in symptoms as assessed by established measures of PTSD, depression, anxiety and perceived stress as a consequence of the experience’ on lines 209-11 and ‘The hypotheses that experimental participants would experience a decrease in symptoms of PTSD, depression, anxiety and perceived stress and an improvement in ratings of general social functioning and psychological growth, relative to controls, were supported.’ On lines 213-16. 

Methods:

I recommend strengthening and clarifying the methods section, as detailed below.

1. Experiment One Design [line 219]: Could the authors please clarify, based on my understanding all 3 group received intervention, there is no control group as such, this design isn’t RCT.

Now lines 233. Yes, you are correct. In study one, all 3 groups received an outdoor recreational intervention (no control group, not an RCT) and all participants took part in only one of the interventions. The between groups factor here relates to the three different intervention experiences (angling vs. equine vs falconry and archery combined). 

2. Random allocation: There could be much more clarity in the Methods about who and what methods where used to generate random allocation to interventions groups.

The manuscript has been updated to include this information: “Participants were randomly allocated (using an online randomisation tool – www.random.org) to either an angling, equine husbandry or falconry and archery combined recreational experience.” (now lines 251-2). Other points of clarification have also been addressed (please see below).

3. Participants: “Eligibility criteria: military veteran with a formal diagnosis of PTSD by a National Health Service or Ministry of Defence psychiatrist. None were currently receiving psychological therapy for PTSD.”

• It isn’t clear what methods and process where used to verify formal diagnosis of PTSD [medical recorders, was it based on self –report?]

We have now added the following to the manuscript: “Diagnosis was confirmed through documents showing a formal diagnosis by a psychiatrist presented by the participants to Veterans First.” (now lines 245-6). See also line 240 introducing Veterans First.

• Given that ‘military veteran’ differs between countries and governments, could the authors please clearly how they define ‘veteran’ in the study?

We have added the following to the manuscript “Definition of the term military veteran differs between countries. In the UK, the term ‘military veteran’ applies to anyone person “who has performed military service for at least one day and drawn a day’s pay” (45; pg.2); however, all our participants had served in the military for considerably longer (mean length of 11 years (SD 6.12). A summary of participant characteristics (including service length) is shown in the left-hand portion of Table 1.” (now lines 246-50).

• There could be much more clarity in the Methods regarding the screen process/data collection, the authors present in Table 1. The summary of participant characteristics [include years’ military service, years since leaving service etc. ]. However, it is not clear how the data was collect. This should have been clearly signposted in the Methods.

 We have now added this information to lines 259-60: “All screening and data collection were carried out via telephone interview by a research assistant.”

4. Intervention descriptions: There could be much more clarity in the Methods about interventions, seating it accrued in each study. Perhaps with the addition of a table which illustrates each intervention group in in each study?

• What was the duration to each session in each intervention group (angling, horse husbandry and riding, falconry and archery)

• What was the activities in each intervention groups? what tools been used how long ? etc..

• How many professional coaches in each intervention group

• What was the training, that professional coaches who provide the intervention had to undertake?

Thank you for this suggestion. We have now included a new table two in the manuscript (and a sentence referring to it – line 270) covering all of these points. On line 265 we also now make clear that each intervention was a day-long experience. 

5. Measurements:

• In the aims [line 193] the authors stated: We designed two experiments to contribute to and extend previous literature by providing an evaluation of the feasibility and effects of brief peer group outdoor recreational”. Could the authors please clarify, how feasibility of the intervention was assessed – please describe how it was done and be how?

Thank you for pointing this issue out. We were using the term ‘feasibility’ in its common usage and not in the clinical trials sense. Therefore, in order to avoid misunderstandings, we have removed this term throughout the manuscript and substituted where appropriate. E.g. the old line 193 has been replaced with “and the potential for military veterans to engage in such interventions” (now line 202). Please also see below.

• To help the reader interpret the effect size, please provide the score range for each of the outcome measure that been used in the study.

Thanks for suggesting this useful addition. We have now added these ranges etc in lines 309-21 and 455-65.

6. Statistical Analyses: Some additional detail and clarification of the data analysis steps used would be helpful. Given the small sample could the authors please add additional information which analysis were used to evaluated inferential analyses and relevant statistical assumptions (including normality, linearity, homoscedasticity).

Small samples of course present challenges in terms of normal distribution. As you will see in the manuscript, we tested for homoscedasticity and when the Mauchley test was significant we applied Greenhouse-Geisser corrections to the relevant F-values. We have amended our manuscript to make that fact more prominent. (lines 33-40 and as previously 347) . Importantly, we have been strict in adhering to the Type I Error Rate (.05) that we set for our analyses. Therefore, you will see that we have not done what some authors do and highlight effects which are “suggestive”, “trend-level” or “approaching significance” - and therefore all effects that we discuss are both large in our sample and statistically significant. This strict/conservative approach mitigates against the inherent difficulties of evaluating statistical assumptions in small samples because it avoids making claims unless the evidence is very clear. Such claims should therefore remain sound even if it has been difficult to evaluate some statistical assumptions. Note, however, that our figures 2 and 4 show the distribution of scores for the most important of our dependent variables, and there is no suggestion there that outliers are driving the effect.

7. The following statement are not fully supported by the results of this study: “The results of the two experiments, comprising four local, outdoor recreational interventions, demonstrate not only the feasibility of motivating veterans with PTSD to engage with such an approach but also its potential clinical usefulness”.

Please see our response to the question 5 above relating to feasibility. We have also replaced that term for ‘potential’ on lines 570 and 588.

8. I am concerned that the authors may be over-interpreting the results of the study. The authors stated in the general discussion: “Another aspect of the experiments reported here is that all participants in both studies had an NHS or Military Physician diagnosis of PTSD. They were followed up to 4 months and data subjected to additional analyses of reliable and clinically significant change. Previous studies of an outdoor experience directed at military veterans have not considered reliable or clinically significant change, relying upon statistical significance that assesses mean difference without reference to the relevance of the effect”.

However:

• It is not clear what was the diagnosis or the severity of PTSD. Potently it could be that the Veteran that agreed to participants were diagnosed with PTSD who is less severe.

This is of course important. The Figures 2 and 4 have now been revised to show the before and after scores on PTSD for every participant who took part in experiment one and experiment two. This enables the reader to examine the PTSD of all participants and to observe the changes obtained in these experiments. We consider that this, together with the open access data ensure transparency. Please see also our response to reviewer one point 2. We have also now added information regarding the diagnosis of PTSD (please see reviewer 2 (yourself) point 3 also. 

• Furthermore, in the analysis it is not clear way the authors used Jacobson and Truax [not in veteran population] nor way they decided to remove segments from the tool. Which led to reduction in the sample to 18.

The PCL-M is among few validated measures of PTSD severity both in line with the DSM-5 and demonstrating excellent psychometric properties. Preliminary cut-scores of both 33 and 38 have been recommended as indicating PTSD presence. These validation efforts were implemented among veteran samples, and optimal PCL cut-scores vary across populations depending on factors like trauma type. It would recommended to use the full valid tool PCL-5 and updated research. Please see: Kazdin AE. Methodological issues and strategies in clinical research. 3rd ed.

The benefits of using the Jacobsen and Traux (1991) method over just relying on cut-off scores on the measures is that this method allows for the percentage of participants achieving a reliable change (as indicated by the Reliable Change Index) to be reported alongside those achieving Clinically Significant Change. This method was similarly used when exploring the psychometric properties of the PCL-5 (Wortmann et al., 2016). The cut-off score of 33 was used within the experiment 2 RCSC analysis as recommended by Wortmann et al. (2016).

---

## [Decision Letter · Decision Letter 1]

9 Oct 2020

PONE-D-19-35181R1

Outdoor recreational activity experiences improve psychological wellbeing of military veterans with post-traumatic stress disorder: positive findings from a pilot study and a randomised controlled trial

PLOS ONE

Dear Dr. Cooper,

Thank you for submitting your manuscript to PLOS ONE. After careful consideration, we feel that it has merit but does not fully meet PLOS ONE’s publication criteria as it currently stands. Therefore, we invite you to submit a revised version of the manuscript that addresses the points raised during the review process.

Overall, your revisions were well received by the reviewers. However, just a couple issues remain. These are straightforward but important and should be addressed.

We look forward to receiving your revised manuscript.

Kind regards,

Ethan Moitra

Academic Editor

PLOS ONE

Brown University

Reviewers' comments:

Reviewer's Responses to Questions

**Comments to the Author**

1. If the authors have adequately addressed your comments raised in a previous round of review and you feel that this manuscript is now acceptable for publication, you may indicate that here to bypass the “Comments to the Author” section, enter your conflict of interest statement in the “Confidential to Editor” section, and submit your "Accept" recommendation.

Reviewer #1: (No Response)

Reviewer #2: (No Response)

2. Is the manuscript technically sound, and do the data support the conclusions?

Reviewer #1: Yes

Reviewer #2: Yes

3. Has the statistical analysis been performed appropriately and rigorously? 

Reviewer #1: Yes

Reviewer #2: Yes

4. Have the authors made all data underlying the findings in their manuscript fully available?

Reviewer #1: Yes

Reviewer #2: Yes

5. Is the manuscript presented in an intelligible fashion and written in standard English?

Reviewer #1: Yes

Reviewer #2: Yes

6. Review Comments to the Author

Reviewer #1: The authors have replied in detail and pertinently to all the reviewer’s points. There are just a couple of my points needing attention:

1. Original line 305 (now 346) Main effect of intervention group: the intervention effect is the interaction group x time, since we are looking for differences between the changes over time among the groups (compare statement of hypotheses in lines 213-215). The main effect includes baseline differences which are not a measure of intervention effect. Thus it should be clearly stated that the effect of the intervention(s) is measured and tested using the group x time interaction.

2. Original line 316 Method for effect sizes. Lenhard and Lenhard is now cited, but the linked web-site provides several alternative definitions of effect size. I could not be sure which one was appropriate, and my attempts to verify this by calculating the effect sized did not yield values in agreement with those in Table 3. Which method was used?

My reservations about the plausibility and conclusiveness of these results remain, but the authors have acknowledged the limitations and future readers should judge for themselves.

Reviewer #2: The authors have provided a comprehensive reply to the questions raised and also presented a greatly improved manuscript. The manuscript is well written, has important clinical message, and should be of great interest to the readers.

7. PLOS authors have the option to publish the peer review history of their article (what does this mean?). If published, this will include your full peer review and any attached files.

Reviewer #1: **Yes: **Jeremy Franklin

Reviewer #2: **Yes: **Neomi Vin-Raviv

---

## [Author Response · Author response to Decision Letter 1]

13 Oct 2020

Dear Professor Moitra,

We thank the reviewers for their review of the manuscript and note that they are both content that the manuscript is technically sound, that the data support the conclusions, that the statistical analysis is appropriate and rigorous, that the data is fully available and that the manuscript is written in clear unambiguous English. 

We also thank you for the opportunity to clarify the two remaining issues raised by Reviewer One. We provide our response below. We believe we have satisfactorily resolved the issues raised by Reviewer One. We have also taken this opportunity to complete a final review and proof read of the manuscript, since we are aware that the journal does not copy edit accepted manuscripts. 

Reviewer #2: The authors have provided a comprehensive reply to the questions raised and also presented a greatly improved manuscript. The manuscript is well written, has important clinical message, and should be of great interest to the readers.

Author Response: We thank Reviewer Two for her positive comments about the manuscript including her observation concerning the importance of the clinical message and likely interest to your wider readership. 

Reviewer #1: The authors have replied in detail and pertinently to all the reviewer’s points. There are just a couple of my points needing attention:

1. Original line 305 (now 346) Main effect of intervention group: the intervention effect is the interaction group x time, since we are looking for differences between the changes over time among the groups (compare statement of hypotheses in lines 213-215). The main effect includes baseline differences which are not a measure of intervention effect. Thus it should be clearly stated that the effect of the intervention(s) is measured and tested using the group x time interaction.

Author Response:

We believe that the reviewer has misunderstood or misread the design of experiment one. We believe that the analyses are correctly reported and interpreted in the current manuscript. The intervention effect is tested differently in experiment one and experiment two. To explain:

The design of experiment one is reported in the abstract (line 39: “experiment one employed a repeated measures design…”) and can also be viewed in Figure One. The statement of hypotheses relevant to experiment one appear on lines 209-211 and state that: “It was hypothesized that participants would experience a decrease in symptoms as assessed by established measures of PTSD, depression, anxiety and perceived stress as a result of the experience.” The reviewer 1 refers to the hypotheses of experiment two on lines 213-215. Experiment two was indeed a controlled experiment and the corresponding hypothesis was that experimental participants would experience a decrease in symptoms relative to controls. We have now made clear that the waitlist-controlled experiment is experiment two in the Aims by adding the words ‘experiment two’ on line 212. 

In fact, in the previous round of reviews, Reviewer 2 (Methods: point one) had also sought clarification that in experiment one, all groups received an outdoor recreational intervention, that is, there was no control group as such, not an RCT. We confirmed that this is the case and that the between groups factor refers to the 3 different types of outdoor activity (lines 235-236). Consequently, from the mixed MANOVA, the important experimental effect that tests our hypothesis is the within-participant effect observed across all activity types as stated on line 347-348. Neither the statistical test of the between groups factor nor the group X time effect in this experiment test our hypothesis. The group X time interaction in this experiment (experiment one) tests whether the changes across time varied according to type of activity (because there is no control group as such). In other words, whether it made any statistical difference if participants experienced angling, or equine or falconry). It turned out that it did not (line 349-351). 

Since the issue here seems to be merely one of stating clearly which is the important test of the hypothesis (as stated on line 343) we have sought to make this even clearer in the text: 

“Participants’ mean psychological wellbeing before the intervention and at 2 weeks and 4 months after for each activity type are summarised in table 3. In order to assess the hypothesised change in psychological wellbeing from before to after the experience, a mixed MANOVA with one between groups factor (intervention type: angling, equine, falconry) and one within-participant factor (3 time points: two weeks pre-intervention, 2 weeks’ post intervention and 4 months post-intervention) was conducted on all four measures of psychological wellbeing. It was hypothesised that the analysis would reveal a significant within-participant effect.”

In experiment two, the reviewer is quite correct that the important intervention effect is the interaction of group x time because this analysis tests whether the change across time differed for the intervention group compared to the control group (see lines 496-7). 

Reviewer 1:

2. Original line 316 Method for effect sizes. Lenhard and Lenhard is now cited, but the linked web-site provides several alternative definitions of effect size. I could not be sure which one was appropriate, and my attempts to verify this by calculating the effect sized did not yield values in agreement with those in Table 3. Which method was used?

Author Response:

Thank you for pointing out that readers may be unclear where to find this information on the cited website. We have now clarified.

The effect sizes shown in Table 3 are for pre-post pairwise comparisons of data. As the reviewer notes there are different methods for different types of data provided on https://www.psychometrica.de/effektstaerke.html

The effect size calculation we used was the one labelled 4: Effect sizes when measurement is repeated within a group (pre-post). We have added detail to the reference no. 56 (lines 816-8) so that together with the available raw data, a reader can calculate the effect sizes should they wish.

56. Lenhard W, Lenhard A. Calculation of effect sizes: Dettelbach: Psychometrica; 2016 [Available from: https://www.psychometrica.de/effektstaerke.html Calculation of Effect Sizes 4. Effect sizes when measurement is repeated within a group (pre-post)]

Finally, we have also run all 4 figures through the PACE image correction software.

Kind regards, Nick Cooper (on behalf of the authors).

---

## [Editor Report · Decision Letter 2]

21 Oct 2020

Outdoor recreational activity experiences improve psychological wellbeing of military veterans with post-traumatic stress disorder: positive findings from a pilot study and a randomised controlled trial

PONE-D-19-35181R2

Dear Dr. Cooper,

We’re pleased to inform you that your manuscript has been judged scientifically suitable for publication and will be formally accepted for publication once it meets all outstanding technical requirements.

Kind regards,

Ethan Moitra

Academic Editor

PLOS ONE
---

## [Editor Report · Acceptance letter]

29 Oct 2020

PONE-D-19-35181R2 

Outdoor recreational activity experiences improve psychological wellbeing of military veterans with post-traumatic stress disorder: positive findings from a pilot study and a randomised controlled trial 

Dear Dr. Cooper:

I'm pleased to inform you that your manuscript has been deemed suitable for publication in PLOS ONE. Congratulations! Your manuscript is now with our production department. 

Kind regards, 

on behalf of

Dr. Ethan Moitra 

Academic Editor

PLOS ONE